# Empirical Testing of a Multidimensional Model of School Dropout Risk

**Borbála Paksi, Krisztián Széll * and Anikó Fehérvári**

Institute of Education, Faculty of Education and Psychology, Eötvös Loránd University, 1075 Budapest, Hungary
*   Correspondence: szell.krisztian@ppk.elte.hu

**Abstract:** Education systems are working to reduce dropout risk, thereby reducing early leaving from education and training rates (ELET) for a more sustainable society. There is a wealth of research on the causes of dropout risk, but little that looks at it in a complex way. Previous research has typically examined the association of a single factor with school dropout. This paper aims to examine the collective relationship between individual, family, and school-level factors and dropout risk based on international literature. Our analyses are based on two online surveys that were conducted among teachers and students in the 2018/2019 and the 2019/2020 academic years respectively (using the data of 2649 students and 2673 teachers from 149 schools in total). Multiple linear regression analyses were performed, and the (ordinary least squares—OLS) regression models were built hierarchically (blockwise entry) with the ENTER method. The research question was which factors are more likely to predict dropout risk. The findings reveal that individual and family factors are far more strongly associated with students' dropout risk than school-related factors. The two strongest individual factors are learning engagement and performance-oriented learning School factors hardly have a role in preventing dropping out of school. Four of the school factors appear to have a definite effect: standards set for students and teachers, belief in the school's role to compensate for disadvantages, and positive school climate. All this draws the attention of practising teachers, school leaders and educational policymakers that the school's protective factors should be stepped up, and the preventive intervention should focus primarily on these factors.

**Keywords:** primary education; early school leaving; ELET; school dropout risk; online survey; complex model

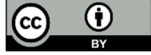

## 1. Introduction

The reduction of early leaving from education and training (ELET) is a key objective for education systems. This is illustrated by the fact that one of the main educational goals of the Europe 2020 strategy is to reduce the average proportion of 18–24-year-olds who have completed at most lower secondary education, that is, whose highest level of education or training attained is ISCED (International Standard Classification of Education) 2011 level 0–2, and are not involved in further education or training below 10% in the European Union (EU)—this is known as the early leavers from education and training statistical indicator. According to Eurostat (2022) data, when Hungary joined the EU in 2004 the average rate of early leavers from education and training in the European Union was higher than in Hungary. Thereafter, the EU average fell steadily, while Hungary's rate moved between 11 and 13%. This, despite the existence, since 2014, of a national strategy to prevent early school leaving and to identify groups at risk of dropping out. In the EU27 the ELET rate was an average of 9.9% in 2020. By contrast, in Hungary, the same rate was 12.1%. Only five member states (Spain, Romania, Italy, Bulgaria, and Malta) have

worse ELET rates than Hungary. However, unlike Hungary, these countries have managed to significantly improve their ELET rates over the past two decades. It is particularly justified to focus on this problem in the context of COVID-19, as the effects and consequences of school closures (Grewenig et al. 2020; Hanushek and Woessmann 2020; Széll et al. 2022) are bound to increase the ELET rate even by the most favourable estimates. It is therefore important to understand the phenomenon and complexity of ELET to prevent it.

Many systematic literature reviews have been published about the factors influencing school dropout, but fewer empirical research projects explored these factors with a complex approach. The diversity of factors associated with school dropout in the literature clearly indicates the complexity of the phenomenon, which can only be captured by complex models. Our paper fills this gap, based on a series of studies that attempted such a complex exploration.

Four systematic literature reviews embracing the complex approach should be highlighted as the basis of our empirical study. Their systematic summary is presented in Table 1. Rumberger (2012) analysed approximately 400 American publications written between 1983 and 2007 and found that the factors indicating a likelihood of dropping out can be classified into two main categories: individual and institutional. The most important individual factors are academic achievement (failure, grade repetition, and performance), behaviour (premature sexual activity, childbirth, drug and alcohol use, criminality, quality of peer relationships, and work), attitudes (beliefs, values, short and long term goals, and self-knowledge), and the socioeconomic background (demography, physical and mental health). Institutional factors include the family (family structure, socioeconomic and educational status, family-school communication, family attitudes and practices), the school (social composition of students, resources, organisational characteristics, and institutional practices), and communities (institutional resources such as child protection and welfare, parents-society engagement, etc.). Lyche (2010) reviewed American and European research in the field and categorised the factors leading to dropout as an individual (related to student and family background), institutional (related to school) or macro level (related to the structure of the educational system or the characteristics of the labour market). Some factors at the school and macro levels can be addressed together, as there are times when the two are not separated, e.g., resources, teacher education, and grade repetition, and there are also macro factors that may underlie individual factors, e.g., working while going to school. Among the individual factors she underlines academic performance as the strongest predictor of dropout, and similarly to Rumberger (2012), she identifies behavioural and background factors, the latter including factors related to family status and attitudes besides the socioeconomic background. In Lyche's overview, the institutional effects only comprise factors that are linked to the school as an institution, specifically, those related to school structure and resources and teaching practices. De Witte et al. (2013) reviewed three decades of literature on school dropout published in English and supplemented the classification of background factors by a social dimension. The review of González-Rodríguez et al. (2019) is a secondary analysis of reviews (i.e., a review of reviews) with a strong new feature of physical and mental health factors (psychology, psychiatry, health science and clinical medicine) added to the sociological and educational aspects of earlier reviews. Another novelty is that the authors do not follow the classification of the other studies mentioned (individual, family, school); instead, they divide the factors into academic and non-academic groups, thus separating the individual's characteristics from school and non-school (demographic and personal) factors.

**Table 1.** Individual, family and school factors of ELET.

| Individual Factors | Family Factors | School Factors |
|---|---|---|
| Background: gender, past experience (preschool, school successes and failures), (physical and mental) health status (well-being), disability, youth pregnancy, working while going to school | Demography, family structure, family type | School type, structure, resources, operator, the composition of students, compensatory role |
| Academic performance, skills and abilities, grade repetition | Attitude to school and learning | Teachers: knowledge, experience, attitudes |
| Behavioural: academic and social engagement (inclusion), deviance, absenteeism | Deviance in the family | School (educational) practices and organisational characteristics: inclusion in the learning process, motivation, school climate, engagement, grade repetition practice, standards, communication between parent and school |
| Belongingness: attributes of peer relations, discrimination, segregation | Socioeconomic and sociocultural status | Communities (institutional resources, e.g., child protection and welfare, parent engagement, social engagement) |

Note: Authors' table based on González-Rodríguez et al. (2019), Lyche (2010), Rumberger (2012) and De Witte et al. (2013).

Some of the systematic reviews not only advocate a comprehensive approach to the problem (Tomaszewska-Pękała et al. 2017) but also argue that some factors are more potent than others: absenteeism, poor academic performance, peer pressure, family structure, economic status, and emotional background have a stronger impact (González-Rodríguez et al. 2019; De Witte et al. 2013).

Besides a similarity in the structure of factors influencing dropout, there is a general agreement in the literature that dropout is the combined effect of several factors. This is difficult to analyse because of its complexity and changes over the course of time static analyses involving only a few of the variables, and two-way tests easily lead the researcher into the pitfall of reinforcing existing stereotypes without demonstrating the combined or interactive effects of the various factors, nor the dynamic nature of their effects (Smeyers 2006). In their systematic review, González-Rodríguez et al. (2019) also conclude that the studies they analysed only address the reasons behind dropout in relation to a group of variables or a specific factor, and there is not enough analysis exploring the interaction between the groups of variables.

No systematic review of the Hungarian empirical studies has been made so far. Narrative reviews suggest that Hungarian dropout research findings have revealed factors similar to those presented in international systematic reviews (Fehérvári 2015; Fehérvári and Tomasz 2015; Paksi et al. 2020). The Hungarian researchers find the same factors contributing to dropout as their international counterparts (see Table 1). Three differences can be discerned in Hungarian studies: (1) The sources of data basically determine what can be analysed and what regular data collections are available for researchers. For instance, in Hungary the connection between teenage pregnancy and dropout is a blank spot—while there are data about women's age when they give birth to their first child, these data cannot be linked to educational databases. By contrast, in other countries, for example in the United States childbirth at school age is a frequently cited dropout factor and is part of the data collection. In addition to researchers' empirical surveys, a large number of Hungarian studies use the data of the Hungarian National Assessment of Basic Competences (NABC), so the indicators in this database largely define the data available for analysis, which is reflected in the ensuing publications. (2) The second difference is while international studies have increasingly adopted an interdisciplinary approach to the dropout problem in recent years, linking different fields of science, in Hungary studies are

dominated by a sociological aspect, and the phenomenon is approached from the angle of the interaction of society and education, specifically inequalities. The social context orientation has long been present in Hungarian educational research: since the 1960s scores of studies have proved that the school perpetuates rather than compensates for social and economic disadvantages, so that students' socioeconomic background accounts for academic performance to a large extent (Ferge 1976). The political changeover and the integrative policies of the 2000s could not change this dominance to any significant or lasting extent. It is conspicuous in the OECD PISA studies: for the past twenty years, Hungary has repeatedly featured among countries where the SES index has the strongest explanatory power in academic achievement (OECD 2019a). The SES index is also relevant in other areas. For example, Hungary's scores are almost similar to the OECD average on indicators of school climate, while the SES index shows a stronger difference from the OECD average on the sense of belonging (OECD 2019b). The educational/pedagogical approach is present, although not markedly, besides the social context but the health and psychological aspect are not typical of the Hungarian studies, and the differences are not linked within a single study. This is connected to the third difference: (3) multiple factor analyses investigating complex causes have been missing from the Hungarian studies.

Over the past two decades, Hungarian education has seen several interventions and innovations aimed at reducing the impact of students' family background on school performance. Thus, integrational and abilities programmes and other school improvements have been implemented, mainly with EU development funds. Research has also shown that the evaluation of these improvements is often neglected or of low quality due to a lack of data. This is underlined by the evaluation of 12 EU programmes for the prevention of early school leaving in the 2017–2021 period. The report also highlights that although the proportion of students at risk of dropping out has decreased and student competencies have increased in schools where interventions have been implemented, spatial disparities persist, and segregation rates have not declined (Molnár and Németh 2022).

In our study, we explore the individual, family and school dimensions and factors of ELET in a Hungarian sample with multivariate analyses visualising the complexity of the problem. The dimensions and factors shown in Table 1 provide the theoretical framework for our analysis.

RQ1: To what extent are individual behaviour and the combination of family and other factors responsible for the risk of dropping out, and how important is each factor in explaining dropout?

RQ2: To what extent is the risk of dropping out explained by a combination of school contextual effects, and which are the significant school contextual effects?

RQ3: To what extent can the risk of dropping out be explained by a combination of individual and school factors, and which of the combination of individual and contextual factors are relevant?

RQ4: How much added value does complex handling of explanatory variables have compared to considering only individual or only school factors?

Our analysis is not only complex in the sense that the models attempt to capture the combined roles of micro and school-level factors but also because of the interdisciplinary (psychological, sociological and educational) aspects of these factors.

## 2. Materials and Methods

Our analyses are based on a research project consisting of a series of cross-sectional studies. Surveys were conducted among both teachers and students in the school participating in the project in two stages, in the 2018/2019 and the 2019/2020 academic years respectively. The analyses discussed in this paper were based on the database linking the student and teacher surveys of both stages. By means of the institutional ID codes, we linked the individual-level student responses and the values of the variables relevant to our topic from the teachers' databases, aggregated by institution, as well as some statistical



data on public education institutions from the mandatory national statistics, and data from the NABC.

### 2.1. Target Populations and Samples

Introduced nationwide in 2016 (Government Decree 229/2012 (VIII. 28.) on the Implementation of the Act on National Public Education) the early warning system collects data on students at dropout risk so that schools can plan intervention and support based on the data. The Hungarian early warning system focuses on ISCED 2 (lower secondary education) and ISCED 3 (upper secondary education). The early warning system generates an index for grades 5–8 primary school students (ISCED 2) based on one or more elementary variables, which shows the proportion of at-risk students at the school level. The schools participating in the project were selected primarily on the basis of data from the early warning system of students at dropout risk, with an effort to optimise the numbers of teachers in five of the 19 counties of Hungary (Vas, Zala, Győr-Moson-Sopron, Borsod-Abaúj-Zemplén and Veszprém Counties) and Budapest. Different schools were surveyed at each stage. The target populations were the grade 7 students and teachers of the selected schools, i.e., 3469 students and 2620 teachers from 83 schools in the 2018/2019 academic year as well as 7342 students and 5968 teachers from 205 schools in the 2019/2020 academic year. There were several reasons for choosing grade 7 students. First, the development project under which the research was carried out focused on ISCED level 2, and grade 7 students are at ISCED level 2 in Hungary. Second, the questionnaire items are best suited to this cohort. And third, grade 7 can be linked to the previous academic year's, i.e., grade 6, data in the NABC. Efforts were made to include the whole of the target groups in the survey.

There were 1953 students in the first stage of the survey (response rate: 56.3%) and 4674 in the second (response rate: 63.7%); the student database thus contains answers from 6627 grade 7 students from 232 schools. The number of teachers surveyed in the first stage was 1136 (response rate: 43.4%) and in the second stage, 2656 (response rate: 44.5%); the teacher database contains answers from 3792 teachers from 267 schools. Since our analysis is based on the linked student and teacher database, only those schools were included in the analysis where both student and teacher surveys were conducted and, for more robust results, only where at least 20% of the students and teaching staff completed the questionnaire. Another criterion was that only those students were included who replied to all of the questions related to the 28 individual student explanatory variables of the model, and for whose schools the 23 contextual (aggregate) variables in our complex model could be composed based on the student and teacher replies. Part of the contextual characteristics of schools was examined based on the aggregated responses of 2673 teachers, and another part was based on the aggregated responses of 2649 students. In our analysis, teacher responses were only used in an aggregate form to construct contextual variables. So, this paper presents analyses of 2649 students from 149 schools including not only the individual characteristics of students but also the contextual characteristics of their schools.

### 2.2. Survey Method and Tools

Self-reporting CAWI surveying technique was used in both target groups (grade 7 students and teachers) and at each stage. The online surveys were conducted in autumn 2018 and autumn 2019 among the institutions participating in the given wave of the project. First, the head teachers were invited to join the research and share the link to the questionnaire with students and teachers in their respective schools. In those schools that agreed to participate, written permission was requested from the parents of the students to complete the questionnaire. The students completed the questionnaire in class. The student questionnaire consisted of 62 questions and the teacher questionnaire of 49 questions.

### 2.3. Ethics

The questionnaires were administered with prior ethics permission granted by the research organisation, ensuring voluntary participation and anonymity. In schools that agreed to participate, informed consent was obtained from students and their parents, in addition to the information and consent of the school and their teachers.

### 2.4. Outcome (Dependent) Variable

The outcome (dependent) variable in our models is an indirect and inverse individual-level indicator of dropout risk: the previous year's GPA. In our analyses, the dropout index calculated for the school is considered to be the reference indicator of dropout which shows the proportion of students at risk of dropping out in the particular school. Pursuant to Section 4, Point 37 of Act CXC of 2011 on National Public Education, the student whose grade point average (GPA) in the given academic year is below satisfactory (3.00) or whose GPA dropped by at least 1.1 compared to the previous academic year is considered an at-risk student and shall require complex systemic educational measures. In the individual analyses, including those presented in this paper, from among the different indicators of academic achievement, which is the strongest predictor of dropout according to the literature (Lyche 2010), we chose as the outcome variable the one whose aggregated mean for the school is most strongly correlated with the grade 7 dropout index. The dropout index shows the percentage of grade 7 students at risk of dropping out compared to the total number of grade 7 students in the particular school. As seen in Table 2, the correlation between the dropout index and the available individual academic achievement indicators aggregated by school is significant. The correlation is strongest ($r = -0.571$) with the year-end GPA of the previous academic year, which is one of the elementary variables in creating the dropout variable.

**Table 2.** Correlation between the dropout index of Grade 7 and the individual academic achievement indicators aggregated by school.

|  | *N* | *r* | *p* |
|---|---|---|---|
| GPA at the end of Grade 6 [a] | 232 | −0.571 | <0.001 |
| Hungarian grammar and literature grade at the end of Grade 6 [a] | 231 | −0.417 | <0.001 |
| Mathematics grade at the end of Grade 6 [a] | 231 | −0.403 | <0.001 |
| History grade at the end of Grade 6 [a] | 231 | −0.394 | <0.001 |
| Rate of students who failed any time [b] | 232 | 0.354 | <0.001 |
| Foreign language grade at the end of Grade 6 [a] | 228 | −0.352 | <0.001 |
| Rate of students who failed at the end of the first-semester [b] | 232 | 0.231 | <0.001 |
| Rate of grade repeating students [b] | 232 | 0.228 | <0.001 |
| Rate of students who failed at the end of a grade but passed on resit [b] | 232 | 0.195 | 0.003 |
| Student satisfaction with school performance [a] | 232 | −0.184 | 0.005 |

Note: Dropout index of Grade 7: Percentage of Grade 7 students at risk of dropping out to the total number of Grade 7 students in the particular school. *N* = number of schools. *r* = Pearson's correlation coefficients. [a] Mean. [b] Percentage (%).

### 2.5. Student Level: Individual and Family (Independent) Variables

Based on the student responses, the four main dimensions of individual factors are captured by 28 variables in total.

The individual background dimension comprises the student's gender, and physical and mental health (well-being). Physical health is measured, on the one hand, by students' self-assessment of their state of health (Cavallo et al. 2015), and on the other hand, by medically established chronic non-infectious disease (by responses to the question 'Does the student have a permanent disease, disability or health condition (diabetes, juvenile arthritis, allergies, birth injury, etc.), diagnosed by a doctor?' 0 = none; 1 = yes). Mental health and well-being were captured by two indicators: the Cantril ladder (Cantril 1965)

measuring satisfaction with life (0 = worst possible life; 10 = best possible life), and Rosenberg's self-esteem scale (Rosenberg 1965).

Within the individual behaviour dimension three variables/indexes measure learning engagement: the question 'How much do you like your favourite subject?' (1 = not at all; 5 = very much); the usefulness of the school/learning index (an index created by principal component analysis of five variables); and the performance-oriented learning index (an index created by principal component analysis of four variables). The content of the principal components can be found in Appendix A. The dimension also includes the variables concerned by deviance created by clustering smoking, drinking, drunkenness, abusive behaviour, abuse suffered (not concerned/concerned), and absenteeism (number of days in an average month when the student skived off, simply didn't go to school).

The belonging to community dimension is measured by the subjective indicator of belonging to the class community ('How much does the student feel part of the class community?' 1 = totally outsider; 5 = integrated) (Kótyuk et al. 2021).

The family dimension contains the family's residence, structure, attitude to the school, and indicators relating to family status and deviance. The level of urbanisation of the residence was depicted by four dummy indicators (capital, county seat, other town, and village). The family structure had six dummy variables (nuclear, stepfamily, single parent, other family, institutionalised), and the number of siblings. The family's attitude to school and learning was measured by means of a four-grade Likert scale ('Do you talk about school life and matters at home?' 1 = yes, regularly; 4 = they are not really interested); patterns of deviance in the family was measured by the number of deviant/risk behaviours that occur in the immediate family (smoking, regular drinking, family member who had been in prison, and use of tranquilisers/sleeping medication). The family's socioeconomic and sociocultural status was depicted by means of five different variables. One is the deprivation index (Fusco et al. 2010; Townsend 1979), which expresses the number of living condition components that are missing for socioeconomic reasons on the basis of four such components (own room, desktop computer/laptop/tablet, internet access at home, at least one week of family holiday per year). Furthermore, the family's sociocultural status was also analysed (the mother and/or the father has/does not have a secondary school leaving certificate); stability of position in the labour market (the mother and/or the father has a permanent job or works in their own enterprise, i.e., whether their status in the labour market is unstable/stable); health status (whether there are any members in the immediate family who is permanently ill: no/yes); and ethnicity (the mother and/or the father is of Roma ethnicity: 0 = no; 1 = yes).

### 2.6. School Level: Contextual (Independent) Variables

The five dimensions of the school context were measured with 23 variables/indexes aggregated by school. All variables/indexes from surveys were aggregated by calculating school-level averages.

The composition of students dimension is measured by the student composition index in the NABC database (which aggregates the rates of students living in above-the-average, and very poor socioeconomic conditions receiving different kinds of support), and based on the student responses, the school-level aggregated rates of students concerned by deviance (smoking, drinking, drunkenness, abusive behaviour, abuse suffered).

Based on the teacher responses, the compensatory role of the school dimension is measured by the compensatory ability index which indicates the extent to which the school can compensate for social disadvantages (created by principal component analysis of four variables), by the segregation support index which indicates a preference for separate education for Roma children (created by principal component analysis of three variables), by the index expressing the family background being a barrier to school socialisation (created by principal component analysis of four variables), and by the index measuring the compensatory role of the individual (e.g., effort) and the family background (e.g., lifestyle and culture) (the index is created by principal component analysis of five

variables). The content of the principal components can be found in Appendix B. Another indicator involved in this dimension from the database of the NABC was whether the school provided integrational/abilities development sessions (0 = no; 1 = yes).

The teachers' dimension contains indicators of teachers' knowledge and experience, and their attitudes regarding the causes of dropout based on the teacher responses aggregated by school. The digital competence variable was developed based on the answers regarding education technology support to teaching and learning (1 = not at all prepared; 5 = fully prepared). The teachers' experience variables in the model represent teachers' acquired experience (average number of hours of continuing training the school's teachers participated in; rate of teachers holding a degree or a PhD in education in the school; the average number of years in service as teachers). Teachers' attitudes regarding the causes of dropout were included by four indexes: (1) the role of the student's individual attributes index (created by principal component analysis of four variables); (2) the role of the student's family background index (created by principal component analysis of four variables); (3) the role of educational and pedagogical factors index (created by principal component analysis of six variables); (4) the role of organisational factors index (created by principal component analysis of four variables). The content of the principal components can be found in Appendix B.

The school (educational) practices and institutional goals dimension contains indicators based on the student and teacher responses aggregated by school. The indicators based on the student responses are the students' climate index representing students' perception of the school climate (created by principal component analysis of 10 variables measured on a scale of 1 = strongly disagree to 4 = strongly agree; variance explained: 48.5%; Cronbach's $\alpha$ = 0.880), the school standards (created by principal component analysis of three variables), and two indexes summarising school goals: the attention paid to students index (created by principal component analysis of three variables), and the preparation for outcomes index (created by principal component analysis of three variables). The indicators of this dimension relying on the teacher responses include the teachers' climate index representing teachers' perception of the school climate (created by principal component analysis of 15 variables); the other two variables describe parent–teacher communication ('How important do you think it is to have good relationships with parents?' 1 = not at all important, 4 = very important; 'What is your relationship with parents?': 1 = decidedly superficial, 4 = decidedly close). The content of the principal components can be found in Appendix B.

The communities (institutional resources) dimension is captured by the teachers' networking index based on the teacher responses (created by principal component analysis of five variables). The content of the principal component can be found in Appendix B.

### 2.7. Analysis Procedure

In the course of our analyses to explain the dropout risk, the variables described above based on the factors of the theoretical model summarised in Table 1 were incorporated into models. The models represent 28 individual and 23 contextual variables and the survey stage variable, altogether an asset of 52 variables. From the variable groups, multiple linear (ordinary least squares—OLS) regression models were built hierarchically (blockwise entry) with the ENTER method (Field 2013), entering the variables blockwise in two different ways, varying the entry order of the blocks of individual and contextual variables. To meet the conditions of multicollinearity, where the effect size was large (above 0.5) between the variables and the level of measurement so allowed, principal component analysis is used to create merged variables (indexes) with 0 mean and 1 standard deviation. In the case of some nominal variables (e.g., parents' labour market status and educational attainment), a logical combination was applied. Variance inflation factor (VIF) is used to detect the severity of multicollinearity in our regression analyses. The VIF is below 3 in all cases.

Two two-stage models were constructed, with the individual and contextual variable blocks entered in different orders. This procedure generates a total of three output models: an individual model, a contextual model, and a complex model, each of which is built on the same set of respondents (2649 students from 149 schools). To refine the interpretation of connections we also present the correlation between the GPA of the previous academic year and the explanatory variables in pairs, without controlling the effect of other variables.

I. The *individual model* investigates to what extent family and other background variables account for the individual risk of dropping out.

II. The *contextual model* investigates the independent role of school factors (compensation for disadvantages, educational practices and organisational characteristics, the composition of students', and teachers' characteristics) in the individual risks of dropping out.

III. The *complex model* attempts to interdisciplinary and multidimensional capture the school-related and individual factors associated with dropping out, and to identify the added explanatory power of the individual and contextual variables.

All analyses were performed with SPSS 26 (IBM Corp. Released 2019. IBM SPSS Statistics for Windows, Version 26.0. IBM Corp., Armonk, NY, USA).

## 3. Results

### 3.1. Individual Level Model

The model we constructed with the set of 29 variables (28 individual characteristics +1 survey stage) is significant ($F_{(27, 2621)} = 46.713$, $p < 0.001$), of medium strength, accounting for approximately one-third of the variance of the GPA (adjusted $R^2 = 0.318$). Out of 19 significant ($p < 0.05$) variables of the model five are related to individual behaviour, another five to the student's background, and nine to the family background, primarily family structure and socioeconomic status. The strongest impact in the model is attributed to factors related to individual behaviour and background: liking the favourite subject ($\beta = 0.222$), performance-oriented learning ($\beta = 0.168$), the respondent's gender ($\beta = 0.139$), and absenteeism ($\beta = -0.118$), as well as the variable representing the family's sociocultural status ($\beta = 0.114$). Correlation with the other 12 significant factors is weaker ($\beta < 0.1$) (see Table 3, individual model).

**Table 3.** Prediction of individual-level risk of dropping out through individual and school factors in two steps, in pairs and complex linear regression models constructed with multiple variable entry.

| | Models by Pairs | | Individual Model (N = 2649) | | Contextual Model (N = 2649) | | Complex Model (N = 2649) | |
|---|---|---|---|---|---|---|---|---|
| | $\beta$ | $p$ | $\beta$ | $p$ | $\beta$ | $p$ | $\beta$ | $p$ |
| Survey stage | 0.087 | <0.001 | 0.047 | 0.004 | 0.066 | 0.001 | 0.035 | 0.046 |
| School level variables | | | | | | | | |
| Composition of students | | | | | | | | |
| School's student composition index | 0.234 | <0.001 | | | 0.058 | 0.022 | −0.041 | 0.072 |
| Proportion of students concerned by deviance | −0.171 | <0.001 | | | −0.093 | <0.001 | −0.016 | 0.397 |
| Compensatory role of the school | | | | | | | | |
| School's compensatory ability index [a] | −0.004 | 0.754 | | | 0.003 | 0.891 | 0.012 | 0.561 |
| Supporting segregation index [a] | 0.018 | 0.183 | | | 0.075 | 0.003 | 0.026 | 0.246 |
| Role of family background in school level socialisation index [a] | −0.034 | 0.010 | | | −0.025 | 0.327 | −0.059 | 0.006 |
| Compensatory role of individual and family background index [a] | −0.065 | <0.001 | | | 0.041 | 0.101 | 0.035 | 0.099 |

| | | | | | | | | |
|---|---|---|---|---|---|---|---|---|
| Does the school provide integrational/abilities development sessions (0 = no; 1 = yes) | 0.003 | 0.841 | | | 0.008 | 0.705 | 0.032 | 0.085 |
| Teachers | | | | | | | | |
| Knowledge: Digital competence (1 = not at all prepared; 5 = fully prepared) | −0.021 | 0.111 | | | −0.027 | 0.281 | −0.014 | 0.503 |
| Experience: Average number of hours of continuing training the school's teachers participated in | −0.096 | <0.001 | | | −0.039 | 0.112 | −0.054 | 0.011 |
| Experience: Proportion of the school's university degree or PhD holder teachers | 0.067 | <0.001 | | | 0.052 | 0.021 | 0.029 | 0.134 |
| Experience: Average number of years the school's teachers spent in service as teachers | 0.025 | 0.048 | | | 0.027 | 0.204 | 0.031 | 0.090 |
| Attitudes regarding the causes of dropout: Role of the student's individual attributes index [a] | −0.124 | <0.001 | | | −0.059 | 0.024 | −0.038 | 0.086 |
| Attitudes regarding the causes of dropout: Role of the student's family background index [a] | −0.034 | 0.009 | | | −0.043 | 0.075 | −0.028 | 0.184 |
| Attitudes regarding the causes of dropout: Role of educational factors index [a] | 0.000 | 0.986 | | | 0.053 | 0.044 | 0.039 | 0.084 |
| Attitudes regarding the causes of dropout: Role of organisational factors index [a] | 0.015 | 0.249 | | | −0.002 | 0.939 | −0.005 | 0.819 |
| School (educational) practices and organisational goals | | | | | | | | |
| Students' climate index [a] | −0.043 | 0.001 | | | −0.024 | 0.445 | −0.031 | 0.244 |
| Teachers' climate index [a] | 0.074 | <0.001 | | | 0.038 | 0.126 | 0.055 | 0.010 |
| Standards: the school's high standards and expectations index [a] | 0.153 | <0.001 | | | 0.132 | <0.001 | 0.099 | <0.001 |
| Parent–teacher communication: How important do you think it is to have good relationships with parents? (1 = not at all important; 4 = very important) | 0.027 | 0.031 | | | −0.020 | 0.431 | 0.008 | 0.698 |
| Parent–teacher communication: What is your relationship with parents? (1 = decidedly superficial; 4 = decidedly close) | 0.060 | <0.001 | | | 0.003 | 0.882 | 0.001 | 0.947 |
| Institutional goals: Attention paid to students' index [a] | 0.036 | 0.003 | | | 0.002 | 0.958 | −0.023 | 0.477 |
| Institutional goals: Preparation for outcomes index [a] | −0.019 | 0.118 | | | −0.067 | 0.068 | −0.046 | 0.152 |
| Communities (institutional resources) | | | | | | | | |
| Closeness of teachers' networking index [a] | −0.047 | <0.001 | | | −0.012 | 0.623 | 0.024 | 0.267 |
| Student level variables | | | | | | | | |
| Individual background | | | | | | | | |
| Respondent's gender (1 = male; 2 = female) | 0.127 | <0.001 | 0.139 | <0.001 | | | 0.129 | <0.001 |
| Physical health: Self–assessment of health status (1 = poor; 4 = excellent) | 0.214 | <0.001 | 0.078 | <0.001 | | | 0.067 | <0.001 |
| Physical health: Chronic non–infectious disease diagnosed by a doctor (0 = none; 1 = yes) | 0.032 | <0.001 | 0.049 | <0.003 | | | 0.042 | 0.012 |
| Mental health and well-being: Satisfaction with life (0 = worst possible life; 10 = best possible life) | 0.248 | <0.001 | 0.065 | 0.001 | | | 0.066 | 0.001 |
| Mental health and well-being: Total score on Rosenberg's self–esteem scale | 0.217 | <0.001 | 0.091 | <0.001 | | | 0.089 | <0.001 |
| Individual behaviour | | | | | | | | |
| Learning engagement: How much do you like your favourite subject? (1 = not at all; 5 = very much) | 0.371 | <0.001 | 0.222 | <0.001 | | | 0.221 | <0.001 |
| Learning engagement: Usefulness of school/learning index [a] | 0.111 | <0.001 | −0.050 | 0.005 | | | −0.044 | <0.016 |

| | β | p | β | p | | β | p |
|---|---|---|---|---|---|---|---|
| Learning engagement: Performance-oriented learning index [a] | 0.347 | <0.001 | 0.168 | <0.001 | | 0.170 | <0.001 |
| Concern by deviance (0 = not concerned; 1 = concerned) | −0.184 | <0.001 | −0.061 | 0.001 | | −0.056 | 0.001 |
| Absenteeism (number of absent days) | −0.267 | <0.001 | −0.118 | <0.001 | | −0.120 | <0.001 |
| **Belonging to community** | | | | | | | |
| Peer relations/segregation: How much does the student feel part of the class community? 1 = totally outsider; 5 = integrated) | 0.117 | <0.001 | −0.004 | 0.816 | | −0.001 | 0.952 |
| **Family background** | | | | | | | |
| Family structure: Nuclear family (0 = no; 1 = yes) [b] | 0.166 | <0.001 | | | | | |
| Family structure: Stepfamily (0 = no; 1 = yes) | −0.066 | <0.001 | −0.048 | 0.004 | | −0.046 | 0.005 |
| Family structure: Singe parent family (0 = no; 1 = yes) | −0.062 | <0.001 | −0.060 | <0.001 | | −0.063 | <0.001 |
| Family structure: Other family (0 = no; 1 = yes) | −0.081 | <0.001 | −0.025 | 0.130 | | −0.025 | 0.131 |
| Family structure: Lives in institution (0 = no; 1 = yes) | −0.106 | <0.001 | −0.041 | 0.014 | | −0.041 | 0.014 |
| Family structure: Number of siblings | −0.230 | <0.001 | −0.049 | 0.004 | | −0.044 | 0.010 |
| Attitude: Do you talk about school life and matters at home? (1 = yes, regularly; 4 = they are not really interested) | −0.147 | <0.001 | 0.019 | 0.276 | | −0.012 | 0.497 |
| Residence: Budapest (0 = no; 1 = yes) [b] | 0.038 | 0.002 | | | | − | − |
| Residence: county seat (0 = no; 1 = yes) | 0.074 | <0.001 | 0.038 | 0.032 | | 0.031 | 0.122 |
| Residence: other town (0 = no; 1 = yes) | −0.021 | 0.083 | −0.012 | 0.527 | | 0.010 | 0.663 |
| Residence: village (0 = no; 1 = yes) | −0.073 | <0.001 | 0.023 | 0.244 | | 0.037 | 0.124 |
| Status: Deprivation index | −0.161 | <0.001 | 0.020 | 0.236 | | 0.015 | 0.366 |
| Status: Does the mother and/or the father have secondary school qualifications? (0 = no; 1 = yes) | 0.297 | <0.001 | 0.114 | <0.001 | | 0.109 | <0.001 |
| Status: Does the father and/or mother have a stable status in the labour market? (0 = unstable; 1 = stable) | 0.088 | <0.001 | 0.054 | 0.001 | | 0.048 | 0.004 |
| Status: Is there any permanent illness in the family (among the parents or siblings)? (0 = no; 1 = yes) | −0.065 | <0.001 | −0.028 | 0.094 | | −0.027 | 0.097 |
| Status: Is the mother and/or father of Roma ethnicity? (0 = no; 1 = yes) | −0.208 | <0.001 | −0.091 | <0.001 | | −0.083 | <0.001 |
| Deviant patterns in the family | −0.159 | <0.001 | −0.044 | 0.009 | | −0.043 | 0.009 |
| Adjusted $R^2$ | | | 0.318 | | 0.070 | 0.331 | |
| $F$ (p) | | | 46.713 (<0.001) | | 9.298 (<0.001) | 27.219 (<0.001) | |

Note: Dependent variable: GPA at the end of the previous academic year. $β$ = Standardized Coefficients. $F$ (p) = Statistics of one-way ANOVA test (significance associated with F statistic). [a] In the case of indexes generated by principal component analysis positive values indicate stronger acceptance of the content of the dimension (i.e., stronger agreement with the statements making up the index). [b] Variable dropped from the model because of the use of dummies.

### 3.2. Contextual Model

The model we constructed with the set of 24 variables (23 contextual characteristics +1 survey stage) is significant ($F(24, 2624) = 9.298$, $p < 0.001$), weak, accounting for approximately 7% of the variance of the GPA (adjusted $R^2 = 0.070$). Similarly to the individual model, the year of the survey is significant in the contextual model too. Of the other seven significant variables ($p < 0.05$) three are related to the teachers' characteristics, two to the composition of students, and one each to the school's compensatory role and the practices adopted by the school. The most powerful indicator in the model is one of the variables related to school practices: the school standards indicator ($β = 0.132$). Less powerful but still strong compared to the other significant variables is the proportion of students concerned by deviance ($β = −0.093$), and teachers' attitudes supporting segregation ($β = 0.075$).

Of the other four variables that are significant but less powerful two are related to teachers' belief that the main reason for dropping out lies in students' characteristics ($\beta$ = −0.059), and in educational factors ($\beta$ = 0.053); and one variable each represents teachers' qualification ($\beta$ = 0.052), and the social composition of students ($\beta$ = 0.058) (see Table 3, contextual model).

### 3.3. Complex Model

With the total set of 50 variables consisting of individual and contextual elements, we managed to construct a model of medium which explains 33.1% of the GPA variance ($F$(50, 2598) = 27.219, $p$ < 0.001, adjusted $R^2$ = 0.331). Incorporating first the 28 individual variables and the year of the survey followed by building in the 23 contextual variables we find that the contextual variables increase the explanatory power of the model consisting of only individual variables (adjusted $R^2$ = 0.318) by merely 1.3 percentage points (33.1–31.8%). However, if we enter the 23 contextual variables and the year of the survey first, the incorporation of variables related to individual behaviour, family and other background factors enhanced the model's explanatory power by 26.1 percentage points (33.1–7.0%) compared to the contextual model constructed only with the set of school-related variables (adjusted $R^2$ = 0.070).

Similarly to the earlier models, the year of the survey proved to have a significant effect on the complex model too. The model presents four significant school-level and 18 significant individual variables ($p$ < 0.05) (see Table 3, complex model).

As regards the number and strength of variables, the complex model is dominated by the individual variables in this respect, too. Eight of these variables represent the students' family background, primarily family structure and family status, five variables are related to student behaviour, and another five to the individual's background. Just as in our model constructed purely from individual variables, individual attitudes have the most prominent influence in the complex model too: love of the favourite subject ($\beta$ = 0.221), performance-oriented learning ($\beta$ = 0.170), the respondent's gender ($\beta$ = 0.129), absenteeism ($\beta$ = −0.120), and the parents' sociocultural status ($\beta$ = 0.102). Correlation with the other 13 significant factors is weaker ($\beta$ < 0.1).

Of the significant school-level variables two are related to school practices, one to the school's compensatory role, and one to teachers' characteristics. In the complex model among the significant school-related variables practices related to high standards should be highlighted as most powerful, but there is no marked hierarchy in the strength of other school variables, the standardised coefficients tend to be around 0.05.

### 4. Discussion

In our study, we attempted a complex analysis of early school leaving. Our goal was to identify the school characteristics that may influence dropout besides students' individual and family-related characteristics. The research framework was developed on the basis of international systematic reviews. We analysed the causes of dropouts in the Hungarian public education system in the context of a complex study conducted in the autumn of the 2018/2019 and the 2019/2020 academic years by means of an online survey of students and teachers.

Systematic reviews of the international literature (Lyche 2010; Rumberger 2012; De Witte et al. 2013) point out that dropout can be traced back to factors falling into three categories: individual, family, and school. The statistical model we built focused on these levels. At the individual level the impact of 11 variables, at the family level that of 17 variables, and at the level of the school, the effect of 23 variables was analysed. As the purpose of our analyses was to predict individual-level dropout risk while, at the same time, data on students at risk of dropping out were available only at the level of the year (or school), we selected the individual variable whose mean aggregated by school most strongly correlated with the school level dropout index, which expressed the percentage of students at risk of dropping out. This variable was the student's GPA. After this, we

examined which of the individual, family, and organisational variables explained the evolution of this outcome variable. To this end, we constructed linear regression models using individual and then organisational (contextual) factors, and finally, a complex hierarchical linear regression model was constructed using the factors of both levels.

We used the individual model to explore the strength of correlation between the variables related to individual behaviour, family, and other background factors in our hypothetical model and the previous end-of-year GPA as the indirect individual-level indicator of dropout. Among the individual factors, we explored the effect of gender, physical and mental health, behaviour (learning and social engagement and involvement, deviance, and absenteeism), as well as belonging. The association between dropout and family background was measured through characteristics including family structure, the family's attitude to learning and the school, demography, socioeconomic and sociocultural status, and deviance in the family. Based on our multivariate individual model constructed with the students' individual characteristics and their family and other background factors, we found that while keeping the other individual and family factors under control, stronger learning engagement, better (or perceived better) mental and physical health, and the family's more favourable sociocultural and labour market status contribute to better academic achievement, hence lesser risk of dropping out. In addition, the dropout risk for girls is significantly lower. Conversely, high absenteeism, involvement in deviance, and family structures with a detrimental effect on the entire family, larger families, and lack of resources in the family tend to involve significantly poorer academic achievement and a greater risk of dropping out. Our model constructed with individual-level variables also reveals that among the behavioural factors, while keeping the other individual and family factors under control, the sense of belonging to the school as a community, and among the family factors, health issues in the family, and the family's attitude to the school do not seem to have a significant role in predicting dropout, and in terms of location of residence, the dropout risk seems to be greater only in the case of county seats.

With our contextual model built on the explanatory variables related to the school, we explored to what extent the individual risk of dropping out is affected by the composition of students, the school's compensatory role, the teachers' knowledge, experience and attitudes, and the school's organisational characteristics, educational practices and external networking. Based on the contextual model built on the school variables we found that while keeping other contextual factors under control, in schools with higher standards, where teachers tend to see educational factors in the background of dropouts, a larger proportion of teachers have ideas supporting segregation, and where the composition of students is more favourable and teachers are more highly qualified students' academic achievement is significantly better, and the risk of individual dropout is lower. By contrast, in schools with a higher proportion of students concerned by deviance, where a greater proportion of teachers tend to "blame" students for dropping out academic achievement is significantly poorer, and students' individual dropout risk is higher. Our contextual model also reveals that among the school factors, while keeping the other individual and family factors under control, the school climate, institutional goals, and quality/closeness of connections with parents and other stakeholders outside the school play no significant part in predicting dropout risk, and even the compensatory role of the school appears only in one indicator.

Based on our complex model constructed with the individual, family and school-related factors we found that the incorporation of the contextual variables did not bring about significant changes in the risk factors plotted by the model based solely on individual variables. While keeping the other individual and contextual factors under control, stronger learning engagement, better (or perceived better) mental and physical health, and the family's more favourable sociocultural and labour market status contribute to better academic achievement, hence lesser risk of dropping out. Absenteeism, involvement in deviance, and family structures with a detrimental effect on the entire family, larger families and lack of resources in the family tend to involve significantly poorer academic

achievement and a greater risk of dropping out. In the complex model, too, girls appear to have a significantly lower dropout risk. Besides the individual impacts, a few weaker institutional patterns can be detected: in schools where standards are set higher and teachers perceive a better school climate, students tend to have better GPAs and fewer dropouts.

Our results confirm the findings underscored in the international literature, namely that some factors have a stronger correlation with dropout than others. Individual characteristics are more strongly associated with success at school than other factors. But there is a difference within the set of individual characteristics: gender, learning engagement, performance-oriented learning, and absenteeism appear to be the strongest predictors. Gender differences in early school leaving and dropout rates show that boys are at a higher risk than girls. The causes of early school leaving are different for boys than for girls. Girls are less at risk, but if they drop out (e.g., due to childbirth), this situation is persistent and returning to school is more difficult. Boys are also overrepresented among early school leavers, although the gap between boys and girls narrows when looking at trend data. However, there are some EU countries with higher early school leaving rates for girls (Bulgaria, Romania). In Hungary, the gap between boys and girls is smaller than the EU percentage (0.7 and 3.5) (Eurostat 2022). Better physical and mental health also predicts school success, though less powerfully than the factors mentioned above. Bowers et al. (2013) drew attention to the positive correlation between substance abuse and dropout. Our study investigated several deviant patterns among students including substance abuse; deviance was found to be associated with poorer school performance. International studies highlight absenteeism and poor performance among the individual factors, but also point out the importance of the personal network of friends and peers (Van Acker and Wehby 2000; Bowers et al. 2013; De Witte et al. 2013; Ekstrand 2015; Veiga et al. 2016). This, however, has not been evidenced by our study. The pairs model reveals that a higher degree of integration goes hand in hand with better academic achievement, but this variable is not included in various other models, so it has not been considered in association with other variables.

Besides individual characteristics family background-related factors also have strong explanatory powers. It appears that in this dimension, sociocultural status has the strongest interaction with students' success. This has been predictable as all previous studies affirm the internationally outstanding explanatory power of family background regarding the student's performance. In addition to the family's sociocultural and socioeconomic status, family structure (single-parent families and families with three or more children), and deviance appearing in the family besides the student's deviance also affect school performance. According to international studies, family structure as well as the affective and economic status are crucial (Bowers et al. 2013; De Witte et al. 2013; Ekstrand 2015; Veiga et al. 2016). Our study also draws attention to the interaction of the family's sociocultural status.

Ekstrand (2015) emphasizes that the time has come to allocate schools greater responsibility in exploring the problem of dropouts. In her analysis, she underlines the positive effect of school climate and engagement (in relation to adults). Yet our study indicates that school factors alone and also in the complex model have a very limited impact on the student's academic achievement. It should be noted that high standards set by the school to students and teachers and the role of family background in school socialisation—in other words, the school's compensatory role is associated with better student performance. International studies addressing the issue of disadvantaged students reiterate concerns about low standards as schools and teachers may at least partly rate students' abilities on the basis of their socioeconomic status (Foschi 2000), setting the student on a more unfavourable learning path, and strengthening the risk of dropping out. In addition to all this, our findings also suggest that a favourable school climate perceived by teachers has a positive effect on students' work.

The international literature abounds with studies (Veiga et al. 2016) that support the importance of parents' attitudes and parent–school communication in students' effectiveness. At the same time, our study did not provide evidence of the interaction of other school-related factors such as teachers' knowledge, experience and classroom practices, or school–parent relations and communication.

*Limitations*

Only students who completed all the questions were included in the analysis, which may bias the results (for example, students of lower motivation, concentration, and literacy all at higher dropout risk may have been excluded from the analysis). However, it is important to note that sampling was based on availability, and we analysed schools in particular regions where students at risk of dropping out are overrepresented. Consequently, the results are not generalisable, the findings and conclusions derived from our analyses only refer to the students and schools in the sample analysed. Due to this, neither the non-response nor the reasons for the non-response have been analysed. Nevertheless, we believe that our sample is suitable for exploring interactions, and fits our research goals. Other limitations are the cross-sectional design and the nature of the outcome variable (the end of the previous year's GPA) which is an indirect individual-level indicator of dropout risk, so our models cannot capture the entire spectrum of dropout risk. It is important to highlight that the database analysed containing both individual and school-level variables may be suitable for performing multilevel analyses, so in the future—taking advantage of the possibilities provided by the data structure—it is worthwhile to examine the relationships by building a multilevel model too.

The research could not take into account the availability of school support staff (school psychologist, social worker, and teaching assistant). Furthermore, the analysis did not focus on all areas and factors of ELET (e.g., adverse childhood experiences), however, we believe that we have been able to capture the most important ones directly or indirectly. It is also important to underline that the research did not consider several macro level (systemic) factors (e.g., characteristics of the education system) that may have an impact on individuals and schools, and hence on the dropout rate too.

## 5. Conclusions

The main strength of this study lies in the combined use of data from the teacher and student surveys, so that student data can be interpreted in the context of the school environment. Another strength is the interdisciplinary and multidimensional approach, i.e., the combined use of psychological, sociological and educational aspects and the combined analysis of the groups of variables (individual, family, school factors) associated with dropout. In some respects, the special nature of the sample is an advantage, as it allows us to identify the individual, family and institutional risk factors and most vulnerable groups within a population characterised by a greater dropout risk. This draws attention to the responsibility of the school and how systemic and school-level interventions, in particular the compensatory role, can contribute to reducing dropout.

For practising teachers and school leaders, it is important to stress that dropout is as much the responsibility of the teacher as it is of the student, and high—but not excessive—standards set for their students and themselves, as well as recognition of the compensatory role of the school and teachers and the creation of a positive, favourable learning-teaching climate can have a significant impact on student achievement. Not forgetting that macro-level (systemic) factors also have an impact on individuals and schools. The project was also explicitly designed to make the research usable in practice: participating schools received not only the general results of the research, but also the results for their own schools, which they could use for their own development purposes.

**Author Contributions:** Conceptualization, B.P., K.S. and A.F.; methodology, B.P., K.S. and A.F.; validation, B.P., K.S. and A.F.; formal analysis, B.P.; investigation, B.P., K.S. and A.F.; data curation, B.P.; writing—original draft preparation, B.P., K.S. and A.F.; writing—review and editing, B.P., K.S. and A.F.; visualization, B.P. and K.S.; supervision, A.F.; project administration, A.F.; funding acquisition, A.F. All authors have read and agreed to the published version of the manuscript.

**Funding:** This research was funded by Human Resources Development Operational Programme, Hungary, No. 3.1.2-16-2016-00001; Title: Methodological renewal of public education to reduce early school leaving. The study was supported by the National Research, Development and Innovation Office – NKFIH, PD 138342.

**Institutional Review Board Statement:** The research was approved by the Research Ethics Committee (REC) at the Faculty of Education and Psychology of Eötvös Loránd University (certificate of approval number 2021/383).

**Informed Consent Statement:** Informed consent was obtained from all subjects involved in the study.

**Data Availability Statement:** Data are contained within the article. The data presented in this study are available on request from the corresponding author.

**Conflicts of Interest:** The authors declare no conflict of interest.

## Appendix A. Principal Components at Student Level

| Principal Component | Component Score |
| --- | --- |
| Individual behaviour dimension | |
| Usefulness of school/learning index | |
| (Variance explained: 45.1%; Cronbach's $\alpha$ = 0.693; 1 = strongly disagree to 4 = strongly agree) | |
| School teaches you things that will be useful later. | 0.701 |
| Most of what you learn in school is unnecessary knowledge. | −0.701 |
| School does not really help you prepare for later life | −0.690 |
| Going to school is a waste of time. | −0.678 |
| I get on better in life by learning. | 0.580 |
| Performance-oriented learning index | |
| (Variance explained: 48.5%; Cronbach's $\alpha$ = 0.632; 1 = not at all true of me to 4 = completely true of me) | |
| It is important for me to do well at school. | 0.803 |
| Grades are important for me, especially for secondary/higher education. | 0.691 |
| I regularly do the homework. | 0.676 |
| It increases my motivation in learning when my teachers, classmates and parents recognise my efforts. | 0.600 |

## Appendix B. Principal Components at School Level

| Principal Component | Component Score |
| --- | --- |
| Compensatory role of the school dimension | |
| School's compensatory ability index | |
| (Variance explained: 44.4%; Cronbach's $\alpha$ = 0.673; 1 = strongly disagree to 4 = strongly agree) | |
| Schools can do much to make students from different social backgrounds more accepting of each other. | 0.731 |
| Roma students can achieve good school results with the right teaching methods. | 0.710 |
| Teachers should take maximum account of differences in the social situation of students' families. | 0.682 |
| For children from multiple disadvantaged backgrounds, the socialisation disadvantages of pre-school can optimally be largely compensated by school. | 0.522 |
| Supporting segregation index | |
| (Variance explained: 65.9%; Cronbach's $\alpha$ = 0.739; 1 = strongly disagree to 4 = strongly agree) | |
| All Roma children have the right to be in the same class as non-Roma children. | 0.708 |
| Roma children do better in separate classes at school. | 0.874 |
| Non-Roma children are better off without Roma children in their class. | 0.843 |
| Role of family background in school-level socialisation index | |

| | |
|---|---|
| (Variance explained: 43.4%; Cronbach's $\alpha$ = 0.654; 1 = strongly disagree to 4 = strongly agree) | |
| If the family does not cooperate with the school, education cannot be truly effective. | 0.711 |
| School cannot be expected to make up for what the family missed out on during early childhood socialization. | 0.663 |
| Parents who do not care about their children's education must be brought to their senses. | 0.642 |
| For Roma children, the most important thing is to learn the rules and adapt to the standards of behaviour expected by the school. | 0.616 |
| Compensatory role of individual and family background index | |
| (Variance explained: 54.1%; Cronbach's $\alpha$ = 0.775; 1 = not at all influential to 5 = extremely influential) | |
| Lifestyle of the family. | 0.847 |
| Culture of the family | 0.821 |
| The attitude of parents. | 0.715 |
| The social background of the family. | 0.644 |
| The diligence and attitude of the child. | 0.623 |
| Teacher dimension | |
| Role of the student's individual attributes index | |
| (Variance explained: 43.2%; Cronbach's $\alpha$ = 0.658; 1 = not at all influential to 5 = extremely influential) | |
| The student is not learning enough. | 0.700 |
| The student's skills are not good enough. | 0.671 |
| The student does not like going to school. | 0.630 |
| The student is deviant and aggressive. | 0.626 |
| Role of the student's family background index | |
| (Variance explained: 49.2%; Cronbach's $\alpha$ = 0.650; 1 = not at all influential to 5 = extremely influential) | |
| The student has a language gap. | 0.765 |
| The student arrives at school with a significant gap. | 0.753 |
| The student is not supported at home in his/her learning. | 0.683 |
| No internet and other modern learning tools at home. | 0.591 |
| Role of educational factors index | |
| (Variance explained: 63%; Cronbach's $\alpha$ = 0.882; 1 = not at all influential to 5 = extremely influential) | |
| The student does not get enough feedback. | 0.865 |
| The teacher does not know enough about the student's strengths. | 0.860 |
| The teacher-student relationship is not good. | 0.791 |
| The student is not regularly given individualised tasks in lessons. | 0.780 |
| The student does not have a sense of achievement in lessons. | 0.744 |
| The student does not know the learning objectives. | 0.711 |
| Role of organisational factors index | |
| (Variance explained: 54%; Cronbach's $\alpha$ = 0.713; 1 = not at all influential to 5 = extremely influential) | |
| The school has no specific strategy to prevent dropouts. | 0.795 |
| The knowledge expected by the school is so far from the knowledge that is important for the student. | 0.728 |
| The classes at school are too large. | 0.719 |
| Students at risk of dropping out cannot be taught in separate groups. | 0.694 |
| School (educational) practices and institutional goals dimension | |
| Students climate index | |
| (Variance explained: 48.5%; Cronbach's $\alpha$ = 0.880; 1 = strongly disagree to 4 = strongly agree) | |
| Most teachers think it is important for students to have a good time at school. | 0.774 |
| Most teachers in school are interested in what students say and think. | 0.753 |
| There is generally a good relationship between teachers and students in this school. | 0.744 |
| This school gives students the opportunity to participate in the decisions that affect them. | 0.707 |
| The school has a climate of mutual support. | 0.703 |
| The school's teachers have a common set of values for teaching and learning | 0.700 |
| This school is a safe place for students. | 0.678 |
| If a student needs extra help, the school will provide it. | 0.675 |
| In this school, teachers see parents as partners. | 0.650 |
| In most cases, parents ask teachers for their child's professional pedagogical opinion. | 0.554 |
| Teachers climate index | |

| | |
|---|---|
| (Variance explained: 47.4%; Cronbach's $\alpha$ = 0.918; 1 = strongly disagree to 4 = strongly agree) | |
| Most teachers think it is important for students to have a good time at school. | 0.747 |
| The school has a climate of mutual support. | 0.739 |
| Most teachers in school are interested in what students say and think. | 0.737 |
| In school, teachers regularly discuss their problems and difficulties with teaching and learning. | 0.731 |
| The school's teachers have a common set of values for teaching and learning. | 0.720 |
| In this school, teachers see parents as partners. | 0.717 |
| If a student needs extra help, the school will provide it. | 0.696 |
| In this school, teachers have the opportunity to participate in decisions that affect them. | 0.692 |
| The headmaster always discusses the school's pedagogical objectives with the teaching staff and usually takes their views into account. | 0.691 |
| This school gives students the opportunity to participate in the decisions that affect them. | 0.683 |
| This school is a safe place for students. | 0.672 |
| There is generally a good relationship between teachers and students in this school. | 0.659 |
| High level of cooperation between the school and the local community. | 0.658 |
| The school also provides appropriate opportunities for students to participate in extra-curricular activities. | 0.635 |
| In most cases, parents ask teachers for their child's professional pedagogical opinion. | 0.520 |
| School's high standards and expectations index (Variance explained: 64.5%; Cronbach's $\alpha$ = 0.724; 1 = not at all typical to 5 = very typical) | |
| High expectations of teachers. | 0.816 |
| High quality of teaching. | 0.797 |
| High expectations of students. | 0.796 |
| Attention paid to students index (Variance explained: 65.2%; Cronbach's $\alpha$ = 0.732; 1 = not at all typical to 5 = very typical) | |
| Personal attention to learners. | 0.831 |
| Paying close attention to disadvantaged students. | 0.825 |
| Paying close attention to gifted students. | 0.764 |
| Preparation for outcomes index (Variance explained: 71.2%; Cronbach's $\alpha$ = 0.794; 1 = not at all typical to 5 = very typical) | |
| Preparing for secondary/higher education. | 0.866 |
| Preparing for a career choice. | 0.845 |
| Teaching to learn. | 0.819 |
| Communities (institutional resources) dimension | |
| Closeness of teachers' networking index (Variance explained: 50.6%; Cronbach's $\alpha$ = 0.750; 1 = no relationship to 5 = very close relationship) | |
| Cooperation with secondary schools in the school district. | 0.778 |
| Cooperation with the kindergartens in the school district. | 0.768 |
| Cooperation with other schools and teachers in the school district. | 0.763 |
| Cooperation with travelling teachers from other schools. | 0.625 |
| Cooperation with professional assistants. | 0.600 |

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
