# Peer review of "Empirical Testing of a Multidimensional Model of School Dropout Risk"

_socsci, doi:10.3390/socsci12020050_

Round 1
Reviewer 1 Report
The topic is important and the effort for a thorough analysis is clearly visible. The study is of high quality, however minor corrections and comments need to be added.
The aim of the study was to examine the complex nature of early school leaving (ESL) and to test the comprehensive, multifactorial model. The complexity of the phenomenon necessitates this comprehensive approach.
The abstract is not as strong as it should be. It does not clarify what the open question is after previous research, which this research aims to answer. Preliminary assumptions are also missing from the abstract. The analysis is based on two online surveys of teachers and students. It is confusing that the research results presented in the abstract are on the one hand about school success and on the other hand about early school leaving. Are the same factors responsible for success as for failure?
Extensive previous research is presented in a meaningful way based on systematic reviews. Table 1 summarizes previous research in a very clear way. The research questions refer to the methodological advantages of complex analysis, and the effects, and advantages of the combined and complex treatment of explanatory variables. This article looks for the answer to a methodological challenge when trying to capture the combined role of macro-, mezzo-, and micro-level factors at the same time.
The paper presents an analysis of 2,649 students from 149 schools, but we do not know how the availability sampling was achieved. In this case, the generalizability of the results is questionable.
It is an important result that the complex model was able to compare the effects of individual and school level factors. Although the student composition of schools is often blamed for student failure, it seems that individual and family factors influence the chance of dropping out more than school-level ones. The strongest factors are absenteeism, involvement in deviance (e.g. substance abuse), and child neglect due to harmful family structures and high child-parent ratios.
Nevertheless, the study managed to identify four school factors that influence ESL. According to the analysis, a favorable school atmosphere perceived by teachers would increase the school's efficiency, teachers however face serious challenges in some schools. ESL students bring with them such serious problems from their family and personal lives that they certainly require more professional support. Neither the theory nor the analytical model took into account the number of students per school psychologist, teacher assistant, or school social worker in the schools. This should be noted as a limitation of the research.
Author Response
Dear Reviewer,
We are very grateful for all your comments and suggestions.
Our answers can be found in the attachment
Please see the attachment.
Best regards,
Author(s)

Reviewer 2 Report
Thank you for the opportunity to review the manuscript entitled “Empirical testing of a comprehensive model of school dropout”. The study uses a sample of 2,649 students from 149 schools to test individual, family, and school-related predictors of student academic achievement.
The study’s strengths include a timely and interesting topic and a large sample. However, I have several concerns related to key aspects of the study.
· First, the paper requires significant language editing to improve readability. The style is convoluted and difficult to understand. I had to read each paragraph several times to grasp what was meant. Oftentimes, the wording was unclear, for example, I was not sure what “structured reviews” (line 51), or “Lyche’s system” (line 69) were. Long and complicated sentences (e.g. lines 27-32) as well as poor structure contribute to the problem. As for structure, see for example the paragraph on the dependent variable (lines 195-214)—the reader learns what the dependent variable is in the two last sentences of this paragraph.
· Second, the paper’s title is inadequate. The study does not seem to test a theoretical model, it instead verifies predictive power of a large set of variables that are considered predictors of dropout. Second, the dependent variable is not dropping out of school (or not), it is student GPA.
· Having said that, the theoretical part is underdeveloped and it does not sufficiently justify the choice of predictors. For example, it is not clear why teacher digital competence (see Table 3) was included – how are such teacher skills related to dropout? The theoretical part just lists a few groups of factors mentioned in the literature. However, we learn neither about mechanisms behind the link between these (groups of) factors and dropout nor about relationships between predictors, which theoretical models need to do. In most cases, the Authors probably mean correlates or risk and protective factors, not factors that are supposed to have a causal relationship with dropout and jointly influence it. Furthermore, the choice of independent variables should be far more selective. Currently, it seems that the Authors used everything they had in the dataset. Alternatively, they should provide a better justification for their choice.
· Moreover, it is questionable how dropout risk was operationalized in the study. As far as I understand, it is student GPA. Although correlated with dropout, GPA is neither dropout nor dropout risk. Due to using GPA as the dependent variable, the paper pertains to school achievement (and could be rewritten as such).
· Furthermore, the analytical methods used in the paper are inadequate. Despite measuring school-level variables in the models, the Authors chose multiple regression (probably OLS regression - it is not stated clearly in the methods section) to predict GPA. Moreover, the analyses did not account for the clustering of students in schools. The analyses should take into account the multilevel data structure; see for example multilevel regression (Snijders & Bosker, 2012).
I provide further recommendations below.
The methods section needs to be significantly improved concerning clarity and detail.
· Sample (line 155+)
· The description should be rewritten to make it more readable. Please provide more information on how the teacher and student samples were related. It seems that teachers and students participated in two separate studies run in different school years. However, did teachers participating in the study teach students who participated in the study a year later? Without such information, it is difficult to understand the design and indicators. For example, it seems that in some cases teachers reported on families (for example, “the index expressing the family background being a barrier to school socialization”, line 276), but did their report on families of students included in the study? How many teachers were included in the analytical sample?
· It seems that concerning student variables, listwise deletion was used (line 177+). This method of dealing with missing data is not recommended. Full information maximum likelihood estimation is a better option, especially if your models include missing data correlates.
· Dependent variable (line 195+). The paragraph requires major revisions. Despite reading it several times, I am still unsure what the dependent variable is and how it was operationalized in the study. Based on the title of this manuscript, I expected it to be dropping out of school (or not), but it does not seem so. Is the dependent variable student GPA in grade 6 measured at the student level? If is it indeed GPA, this paper is neither about dropout nor about dropout risk, it is about student achievement.
· Independent variables (line 221+)
· This section is difficult to follow. Moreover, it will benefit from providing more detail on the measured variables. For example, it is not clear what such indexes as “the compensatory ability index” or “the segregation support index” measure and how exactly they were created (line 273+). Please summarize information on all your variables in a table and provide a definition for each variable, information on the number of items, item wording for example items; indicate the way of creating each variable (averaging, PCA, CFA, etc.); report response scales, ranges, and reliabilities (if applicable); provide information on validity (if applicable) and informants or data sources (students, teachers, school records, etc.). If necessary, some information may be presented in an online supplement.
· The way that school-level variables (line 266+) are described is unclear and confusing. Some variables that seem to pertain to the family are considered school variables, for example, “the compensatory role of the individual (e.g. effort) and the family background (e.g. lifestyle and culture)”, see line 279. Providing definitions for all variables may solve the problem.
· All school-level variables should be measured at the school level, but it is difficult to say if it is the case. On one hand, the word “aggregated” appears multiple times in the paragraph which suggests that they were created by calculating for example a school-level average. On the other hand, PCA is mentioned multiple times – was it PCA run on teacher- or student-reported variables? If yes, how were the variables aggregated later? How some of the school-level variables were created is also questionable. For example, the index of school climate was created using PCA, but it should be a two-level CFA or EFA, see for example Stapelton et al. (2016), Köhler et al. (2020), or Marsh et al. (2012). For aggregated measures, ICC2 should be provided (see e.g., Bliese, 2000).
· Similarly, the Authors should use factor scores derived from CFA or EFA instead of PCA to create some of the student-level indicators (for example, the usefulness of school/learning or performance-oriented learning) because they are latent variables.
· Analysis procedure (line 346+)
· Information on software that was used is lacking. It is not clear what the ENTER procedure is – is it specific to the software that you used?
· Please provide more detail on multicollinearity and changes to your collinear indicators. Without it is impossible to replicate your analyses.
· As mentioned before, the analyses did not account for the multilevel structure of the data (or at least I did not find evidence that they did, for example, information on fixed and random effects, etc.).
· Results and Discussion
· Please provide a correlation table and descriptive statistics for all of the variables in the study, either in the results section or in an online supplement.
· The discussion section largely echoes the results. Instead, it should discuss the results in light of existing literature and past studies.
· Moreover, it refers repeatedly to dropout and dropout risk, which is not warranted because the dependent variable was GPA.
· The conclusion that school factors play a minor role in predicting dropout (or dropout risk) in comparison to family and individual factors is also unwarranted. First, the dependent variable was neither dropout nor dropout risk, it was GPA. Second, school-level factors were improperly treated as level 1 variables and the analyses did not account for the multilevel structure of the data. Finally, the theoretical part does not present a compelling theoretical model that would indicate school-level factors that are key to dropout. In other words, there is not enough evidence that school-level factors included in the study are the ones that should have been included.
· The discussion section does not discuss the cross-sectional design as an important limitation of this study. Furthermore, no classroom-level variables were included.
Additionally, since the study was run in Hungary it would be useful to have basic information on the Hungarian education system. For example, how many grades do primary and secondary schooling include? At which ISCED level students in Grade 7 are? Without it is difficult to assess if focusing on grade 7 was a good choice.
I hope these comments are helpful to the Authors’ further work.
References
Bliese, P. D. (2000). Within-group agreement, non-independence, and reliability: Implications for data aggregation and analysis. In K. . J. Klein & S. W. Kozlowski (Eds.), Multilevel theory, research, and methods in organizations (pp. 349–381). Jossey-Bass. https://www.kellogg.northwestern.edu/rc/workshops/mlm/Bliese_2000.pdf
Köhler, C., Kuger, S., Naumann, A., & Hartig, J. (2020). Multilevel models for evaluating the effectiveness of teaching. Zeitschrift Für Pädagogik, 66(1), Article 1.
Marsh, H. W., Lüdtke, O., Nagengast, B., Trautwein, U., Morin, A. J. S., Abduljabbar, A. S., & Köller, O. (2012). Classroom climate and contextual effects: Conceptual and methodological issues in the evaluation of group-level effects. Educational Psychologist, 47(2), 106–124. https://doi.org/10.1080/00461520.2012.670488
Snijders, T., & Bosker, R. (2012). Multilevel analysis: An introduction to basic and advanced multilevel modeling (2nd ed). Sage.
Stapleton, L. M., Yang, J. S., & Hancock, G. R. (2016). Construct meaning in multilevel settings. Journal of Educational and Behavioral Statistics, 41(5), 481–520. https://doi.org/10.3102/1076998616646200
Author Response
Response to Reviewer 2 Comments
Point 1: Thank you for the opportunity to review the manuscript entitled “Empirical testing of a comprehensive model of school dropout”. The study uses a sample of 2,649 students from 149 schools to test individual, family, and school-related predictors of student academic achievement.
The study’s strengths include a timely and interesting topic and a large sample. However, I have several concerns related to key aspects of the study.
I hope these comments are helpful to the Authors’ further work.
Response 1: We are very grateful for all your useful comments and suggestions.
Point 2: First, the paper requires significant language editing to improve readability. The style is convoluted and difficult to understand. I had to read each paragraph several times to grasp what was meant. Oftentimes, the wording was unclear, for example, I was not sure what “structured reviews” (line 51), or “Lyche’s system” (line 69) were. Long and complicated sentences (e.g. lines 27-32) as well as poor structure contribute to the problem. As for structure, see for example the paragraph on the dependent variable (lines 195-214)—the reader learns what the dependent variable is in the two last sentences of this paragraph.
Response 2: The language proofreader has also reviewed the text again from this point of view. The term “structured reviews” has been replaced by “systematic reviews” throughout the text. “Lyche’s system” has been replaced by “Lyche’s overview”.
Furthermore, we have changed the structure of the paragraph on the dependent variable, and the following sentence has been moved to the beginning of the paragraph.
«The outcome (dependent) variable in our models is an indirect and inverse individual-level indicator of dropout risk: the previous year's GPA.»
Point 3: Second, the paper’s title is inadequate. The study does not seem to test a theoretical model, it instead verifies predictive power of a large set of variables that are considered predictors of dropout. Second, the dependent variable is not dropping out of school (or not), it is student GPA.
Having said that, the theoretical part is underdeveloped and it does not sufficiently justify the choice of predictors. For example, it is not clear why teacher digital competence (see Table 3) was included – how are such teacher skills related to dropout? The theoretical part just lists a few groups of factors mentioned in the literature. However, we learn neither about mechanisms behind the link between these (groups of) factors and dropout nor about relationships between predictors, which theoretical models need to do. In most cases, the Authors probably mean correlates or risk and protective factors, not factors that are supposed to have a causal relationship with dropout and jointly influence it. Furthermore, the choice of independent variables should be far more selective. Currently, it seems that the Authors used everything they had in the dataset. Alternatively, they should provide a better justification for their choice.
Moreover, it is questionable how dropout risk was operationalized in the study. As far as I understand, it is student GPA. Although correlated with dropout, GPA is neither dropout nor dropout risk. Due to using GPA as the dependent variable, the paper pertains to school achievement (and could be rewritten as such).
Response 3: The title of the study has been changed to “Empirical testing of a multidimensional model of school dropout risk”.
The dimensions and factors shown in Table 1 provide the theoretical framework for our analysis. This has been highlighted more directly in the text. So, Table 1 presents the factors influencing the risk of early school leaving that we have addressed in our analysis. In our opinion, these variables are organised in the system described in Table 1, so that the combination of the variables analysed can be interpreted as a complex model of the risk of early school leaving. For example, in the teachers' dimension, indicators were used to measure teachers' knowledge, experience, and attitudes regarding the causes of dropout. The digital competence variable was used as an indicator of teacher knowledge.
Furthermore, in our view, student GPA is a valid individual-level indicator of dropout risk, and we have also presented this in the Outcome (dependent) variable subsection (2.4) of our paper.
Point 4: Furthermore, the analytical methods used in the paper are inadequate. Despite measuring school-level variables in the models, the Authors chose multiple regression (probably OLS regression - it is not stated clearly in the methods section) to predict GPA. Moreover, the analyses did not account for the clustering of students in schools. The analyses should take into account the multilevel data structure; see for example multilevel regression (Snijders & Bosker, 2012).
Response 4: In our analyses to explain the dropout risk (measured by student GPA), 28 individual and 23 contextual variables based on the theoretical model factors summarised in Table 1 were incorporated into models. Multiple linear regression analyses were performed, the (ordinary least squares—OLS) regression models were built hierarchically (blockwise entry) with the ENTER method (Field 2013). Two two-stage models were constructed, with the individual and contextual variable blocks entered in different orders.
(Field 2013) Field, Andy. 2013. Discovering statistics using IBM SPSS statistics. (4th ed) London: SAGE Publications.
Point 5: I provide further recommendations below.
The methods section needs to be significantly improved concerning clarity and detail.
Sample (line 155+)
The description should be rewritten to make it more readable. Please provide more information on how the teacher and student samples were related. It seems that teachers and students participated in two separate studies run in different school years. However, did teachers participating in the study teach students who participated in the study a year later? Without such information, it is difficult to understand the design and indicators. For example, it seems that in some cases teachers reported on families (for example, “the index expressing the family background being a barrier to school socialization”, line 276), but did their report on families of students included in the study? How many teachers were included in the analytical sample?
It seems that concerning student variables, listwise deletion was used (line 177+). This method of dealing with missing data is not recommended. Full information maximum likelihood estimation is a better option, especially if your models include missing data correlates.
Response 5: Thank you for your important comments and suggestions. The study has been supplemented.
«Surveys were conducted among both teachers and students in the school participating in the project in two stages, in the 2018/2019 and the 2019/2020 academic years respectively. The analyses discussed in this paper were based on the database linking the student and teacher surveys of both stages. By means of the institutional ID codes, we linked the individual-level student responses and the values of the variables relevant to our topic from the teachers databases, aggregated by institution, as well as some statistical data on public education institutions from the mandatory national statistics, and data from the NABC.» (see lines 179-186)
«The target populations were the grade 7 students and teachers of the selected schools, i.e. 3,469 students and 2,620 teachers from 83 schools in the 2018/2019 academic year as well as 7,342 students and 5,968 teachers from 205 schools in the 2019/2020 academic year.» (see lines 198-201)
«There were 1,953 students in the first stage of the survey (response rate: 56.3%) and 4,674 in the second (response rate: 63.7%); the student database thus contains answers of 6,627 grade 7 students from 232 schools. The number of teachers surveyed in the first stage was 1,136 (response rate: 43.4%) and in the second stage, 2,656 (response rate: 44.5%); the teacher database contains answers of 3,792 teachers from 267 schools.» (see lines 208-212)
«Part of the contextual characteristics of schools was examined based on the aggregated responses of 2,673 teachers, and another part was based on the aggregated responses of 2,649 students. In our analysis, teacher responses were only used in an aggregate form to construct contextual variables.» (see lines 219-222)
Furthermore, listwise deletion was actually used in the analyses. On the one hand, the benefits of the imputation of missing data are far from clear in the literature, and on the other hand, our sampling was based on availability, and we analysed schools in particular regions where students at risk of dropping out are overrepresented. Consequently, the results are not generalisable, the findings and conclusions derived from our analyses only refer to the students and schools in the sample analysed. Due to this, neither the non-response nor the reasons for the non-response have been analysed. This information has been added to the Limitations subsection (4.1) of the study.
«However, it is important to note that sampling was based on availability, and we analysed schools in particular regions where students at risk of dropping out are overrepresented. Consequently, the results are not generalisable, the findings and conclusions derived from our analyses only refer to the students and schools in the sample analysed. Due to this, neither the non-response nor the reasons for the non-response have been analysed.»
Point 6: Dependent variable (line 195+). The paragraph requires major revisions. Despite reading it several times, I am still unsure what the dependent variable is and how it was operationalized in the study. Based on the title of this manuscript, I expected it to be dropping out of school (or not), but it does not seem so. Is the dependent variable student GPA in grade 6 measured at the student level? If is it indeed GPA, this paper is neither about dropout nor about dropout risk, it is about student achievement.
Response 6: Please see our Response 3.
Point 7: Independent variables (line 221+)
This section is difficult to follow. Moreover, it will benefit from providing more detail on the measured variables. For example, it is not clear what such indexes as “the compensatory ability index” or “the segregation support index” measure and how exactly they were created (line 273+). Please summarize information on all your variables in a table and provide a definition for each variable, information on the number of items, item wording for example items; indicate the way of creating each variable (averaging, PCA, CFA, etc.); report response scales, ranges, and reliabilities (if applicable); provide information on validity (if applicable) and informants or data sources (students, teachers, school records, etc.). If necessary, some information may be presented in an online supplement.
The way that school-level variables (line 266+) are described is unclear and confusing. Some variables that seem to pertain to the family are considered school variables, for example, “the compensatory role of the individual (e.g. effort) and the family background (e.g. lifestyle and culture)”, see line 279. Providing definitions for all variables may solve the problem.
All school-level variables should be measured at the school level, but it is difficult to say if it is the case. On one hand, the word “aggregated” appears multiple times in the paragraph which suggests that they were created by calculating for example a school-level average. On the other hand, PCA is mentioned multiple times – was it PCA run on teacher- or student-reported variables? If yes, how were the variables aggregated later? How some of the school-level variables were created is also questionable. For example, the index of school climate was created using PCA, but it should be a two-level CFA or EFA, see for example Stapelton et al. (2016), Köhler et al. (2020), or Marsh et al. (2012). For aggregated measures, ICC2 should be provided (see e.g., Bliese, 2000).
Similarly, the Authors should use factor scores derived from CFA or EFA instead of PCA to create some of the student-level indicators (for example, the usefulness of school/learning or performance-oriented learning) because they are latent variables.
Response 7: Based on your comments and suggestions, we have added to these sections of the study (please see 2.5 and 2.6 sections).
The index measuring the compensatory role of the individual (e.g. effort) and the family background (e.g. lifestyle and culture) is based on teacher responses, i.e. it cannot be considered an individual-level variable, but only a school-level variable that indirectly shows the compensatory role of the school.
We did not want to extract latent factors from our dataset but we performed a "simple" data reduction on our data set. When data reduction is the goal the principal component analysis is the appropriate choice (to explain a maximal amount of variance with as few principal components as possible). That's why we used PCA in our analysis.
Point 8: Analysis procedure (line 346+)
Information on software that was used is lacking. It is not clear what the ENTER procedure is – is it specific to the software that you used?
Please provide more detail on multicollinearity and changes to your collinear indicators. Without it is impossible to replicate your analyses.
As mentioned before, the analyses did not account for the multilevel structure of the data (or at least I did not find evidence that they did, for example, information on fixed and random effects, etc.).
Response 8: All analyses were performed with IBM SPSS Statistics for Windows, Version 26.0 (IBM 2019). This additional information has been added to the end of the Analysis Procedure section (2.7).
Forced entry (or ENTER as it is known in SPSS) is a method in which all predictors are forced into the model simultaneously.
There was not multicollinearity, as it was handled by variable and index formation (e.g. PCA) (please see 2.5 and 2.6 sections).
Multiple linear regression analyses were performed, the (ordinary least squares—OLS) regression models were built hierarchically (blockwise entry) with the ENTER method (Field 2013).
(IBM 2019) IBM Corp. Released 2019. IBM SPSS Statistics for Windows, Version 26.0. Armonk, NY: IBM Corp.
(Field 2013) Field, Andy. 2013. Discovering statistics using IBM SPSS statistics. (4th ed) London: SAGE Publications.
Point 9: Results and Discussion
Please provide a correlation table and descriptive statistics for all of the variables in the study, either in the results section or in an online supplement.
The discussion section largely echoes the results. Instead, it should discuss the results in light of existing literature and past studies.
Moreover, it refers repeatedly to dropout and dropout risk, which is not warranted because the dependent variable was GPA.
The conclusion that school factors play a minor role in predicting dropout (or dropout risk) in comparison to family and individual factors is also unwarranted. First, the dependent variable was neither dropout nor dropout risk, it was GPA. Second, school-level factors were improperly treated as level 1 variables and the analyses did not account for the multilevel structure of the data. Finally, the theoretical part does not present a compelling theoretical model that would indicate school-level factors that are key to dropout. In other words, there is not enough evidence that school-level factors included in the study are the ones that should have been included.
The discussion section does not discuss the cross-sectional design as an important limitation of this study. Furthermore, no classroom-level variables were included.
Response 9: More than 150 variables were used for the analysis. For example, 52 variables were used to construct the indexes based on student responses and 54 variables were used to construct the indexes based on teacher responses. A detailed description of these variables would be beyond the scope of this study.
We analysed survey results objectively in the Results section (3) and described findings in the Discussion section (4). Furthermore, there are eight references in the Discussion section (4), several of which are systematic literature reviews.
We have added the following to the Limitations subsection (4.1): «Other limitations are he cross-sectional design and the indirect nature of the outcome variable (end-of-year GPA) as it cannot capture the entire spectrum of dropout risk.»
We believe that we have responded to the other comments in our previous responses.
Point 10: Additionally, since the study was run in Hungary it would be useful to have basic information on the Hungarian education system. For example, how many grades do primary and secondary schooling include? At which ISCED level students in Grade 7 are? Without it is difficult to assess if focusing on grade 7 was a good choice.
Response 10: The Hungarian early warning system focuses on ISCED 2 (lower secondary education) and ISCED 3 (upper secondary education). While the focus is more on prevention at the ISCED 2 level, the focus is more on intervention at the ISCED 3 level. The development project under which the research was carried out focused on ISCED 2 level. Grade 7 students are related to ISCED 2 in Hungary. Among others, this is one of the reasons why we have chosen ISCED level 2 and grade 7 students. This information has been added to the study.
«There were several reasons for choosing grade 7 students. First, the development project under which the research was carried out focused on ISCED level 2, and grade 7 students are at ISCED level 2 in Hungary. Second, the questionnaire items are best suited to this cohort. And third, grade 7 can be linked to the previous academic year’s, i.e., grade 6, data in the NABC.» (see lines 202-206)
Reviewer 3 Report
In general terms it is an interesting article. Below I highlight aspects that, in my opinion, need to be improved.
In the abstract, you refer to the method move to other aspects and back to the method. If you put it all together it will ease readers understanding of your work.
The introduction is sound. However, there seems to be a conceptual misunderstanding revealed by the unselective use of ‘dropout’ and the analysis of ESL, which reports to specific conditions of dropout. This needs to be clarified. Moreover, the concept of ESL has been substituted in political documents and in the literature by the concept of ELET (early leaving from education and training) that shelters the diversity of educational offers. ELET might be a more adequate concept to use in the article.
There are a set of macro level (systemic) factors that go beyond demography and are not considered in the analysis. This is OK but you should at least introduce a paragraph that recognizes the impact of these factors on individuals and schools. This would allow the analysis not to put the 'blame on the victim' – individuals, families and schools.
Concerning the ethical aspects of the surveys it seems that institutional consent is not enough. Individual consent and assent should have been considered. Did you forget to mention? What are the ethical guidelines you are following?
The methods of data collection and analysis are well detailed.
Please revise the dates of the empirical work 2018 to 2020 or 2018-2019 (as referred in the discussion).
Your conclusions seem to fall a bit short and do not cover all the aspects that were analyzed.
Pay more attention to the ‘use of English’. Some spelling mistakes were identified.
Author Response

(The authors gave the same response as above.)

Reviewer 4 Report
There are many strengths in this paper, including the wide range of variables such as mental health and wellbeing of the individual, as well as absenteeism, abuse, drug use focus, together with a range of academic attainment and failure variables. This is combined with school belongingness themes and a family focus on often neglected factors such as sleep medication and having been in prison. So this is clearly a multidimensional approach at individual, school and family levels. It also incorporates a necessary focus on poverty aspects.
This is very far however from a comprehensive review of school drop out/early school leaving literature and the claim must be reformulated to describe it as a multidimensional and interdisciplinary model rather than a comprehensive model. In doing so, the paper needs to be explicit regarding which areas of early school leaving/drop out have not been addressed. For example, a number of ACEs (Adverse Childhood Experiences) have not been included and there is no direct focus on bullying. See also the EU Commission's multidimensional evaluation for a range of other dimensions including system supports https://op.europa.eu/en/publication-detail/-/publication/72f0303e-cf8e-11e9-b4bf-01aa75ed71a1/language-en
The role of culture and society in drop out needs wider recognition to allow for cultural variance. Specific features of the Hungarian societal and educational context need to be described to allow the reader to assess the cultural transferability of factors. For example to situate the school climate, belongingness focus it would be helpful if the PISA scores for Hungary on feeling like an outsider, feeling like belonging in school e.g., in 2012 and after were stated to compare with OECD average.
The conclusion that school factors are less strong needs to recognise the limitation of the study that it was not profiling multidisciplinary team supports in and around schools that are in many European countries, for example, to address absenteeism or mental health. A related point is that absenteeism is treated as an individual factor though it can also be related to the systems of services monitoring absenteeism. Other system profiles in Hungary that would be helpful for the reader to contextualise the meaning of the data include - is there a national early school leaving strategy in Hungary to implement the ET2020 Headline target for Hungary ? How do the poverty rates in Hungary nationally and even regionally compare with other EU countries ? Are hot meals provided in schools ? There is some system profile for example regarding whether the school provided integrational/abilities developmental sessions.
Only those students who completed all the questions were included in the sample - some analysis is needed of the bias in doing this - for example, it may exclude students of lower motivation, concentration, literacy all at higher risk of early school leaving. Furthermore, as it is a school based sample it excludes those students absent from school when the test takes place, again a factor to be noted as a limitation . Other dimensions relevant to early school leaving such as loneliness in school (Frostad et al: Per Frostad, Sip Jan Pijl & Per Egil Mjaavatn (2015) Losing All Interest in School: Social Participation as a Predictor of the Intention to Leave Upper Secondary School Early, Scandinavian Journal of Educational Research, 59:1, 110-122, DOI: 10.1080/00313831.2014.904420 ) and sleep deficits of students https://op.europa.eu/en/publication-detail/-/publication/6e48090a-e204-11e6-ad7c-01aa75ed71a1 as well as bullying and early school leaving https://op.europa.eu/en/publication-detail/-/publication/3fb78afb-c03d-11e6-a6db-01aa75ed71a1 require noting as not being addressed in the study (see EU Commission published reports above) and hence for future expansion of the model or limits to the current model. The discussion notes the gender differences in a Hungarian context and again this needs to be situated comparatively in an EU context (see Donlevy et al 2019) https://op.europa.eu/en/publication-detail/-/publication/72f0303e-cf8e-11e9-b4bf-01aa75ed71a1/language-en. Some theorising of protective factors that are addressed in the study, often as compensatory factors, would be helpful to understand their relation to risk factors.
Author Response

(The authors gave the same response as above.)

Round 2
Reviewer 2 Report
While I appreciate the Authors' efforts to revise this manuscript, I am sorry to say that my comments cannot be more positive at the moment. Although the quality of the manuscript has improved, most of the issues have been addressed superficially and without necessary detail.
1. The study’s theoretical background remains underdeveloped. Although the Authors state that the study tests a model of dropout risk, the theoretical part is limited to a list of several groups of dropout risk predictors (identified in past studies) without any attempt to conceptually integrate them or explain the mechanism behind the relationships. Such a list is not a model.
2. The statistical analyses remain inadequate. They (a) do not account for the clustered structure of the data and (b) test school-level variables at the individual level.
3. I am still in doubt about the use of PCA for creating at least some of the indicators. Still, there is still no correlation table for the indicators in the study. It is key in a study with this number of variables that the readers can assess by themselves whether some of the results may be due to collinearity. Stating that there was not collinearity is not enough.
4. It is very concerning that the Authors did not provide more detailed information on the constructs included in the study (including their definitions) and scales / measurement tools used to operationalize these constructs. Without such basic information it is impossible to (a) understand what factors the study tested, (b) replicate the results.
5. Due to the underdeveloped theoretical part and inadequate data analysis, I do not believe that the conclusions are sufficiently supported.
Author Response
Dear Reviewer,
We are very grateful for all your comments and suggestions.
Our answers can be found in the attachment.
Please see the attachment.
Best regards,
Author(s)
